# Emotionally congruent music and text increase immersion and appraisal

**Pia Hauck** [1,2]*, **Heiko Hecht** [1]

**1** Department of General Experimental Psychology, Johannes Gutenberg-Universität Mainz, Mainz, Germany, **2** Institute for Research on Reading and Media, Stiftung Lesen, Mainz, Germany

* hauckp@uni-mainz.de

**Data Availability Statement:** All relevant data are within the manuscript and its Supporting Information files. They are also uploaded on OSF: DOI 10.17605/OSF.IO/SF47R.

**Competing interests:** The authors have declared that no competing interests exist.

## Abstract

Numerous studies indicate that listening to music and reading are processes that interact in multiple ways. However, these interactions have rarely been explored with regard to the role of emotional mood. In this study, we first conducted two pilot experiments to assess the conveyed emotional mood of four classical music pieces and that of four narrative text excerpts. In the main experiment, participants were asked to read the texts while listening to the music and to rate their emotional state in terms of valence, arousal, and dominance. Subsequently, they rated text and music of the multisensory event in terms of the perceived mood, liking, immersion, and music-text fit. We found a mutual carry-over effect of happy and sad moods from music to text and vice versa. Against our expectations, this effect was not mediated by the valence, arousal, or dominance experienced by the subject. Moreover, we revealed a significant interaction between music mood and text mood. Texts were liked better, they were classified as of better quality, and participants felt more immersed in the text if text mood and music mood corresponded. The role of mood congruence when listening to music while reading should not be ignored and deserves further exploration.

## Introduction

Multitasking is a phenomenon of our time. Smartphones and other digital devices are omni-present in our daily lives. Apart from making phone calls, we send text messages, check the news, read books, make photos, etc. all the while listening to music. Our increasing mobility and the option to be constantly online contribute to a continuous sensory overload. Visual input via the display, auditory signals delivered by headphones, and haptic stimuli like the smartphone vibration merge with sensations from the surrounding world into complex multi-sensory impressions. Reading texts and simultaneously listening to music has become a common combination of activities, sometimes intending to create a musical background for the reading content, sometimes just to seal oneself off from surrounding noise. But what does this new level of intersensory fusion do to our emotions? Both music [1] and text [2] are known to be able to transport distinct emotions. But what happens to our emotion recognition when we are consuming music and text at the same time? Does the music influence our impression of the mood expressed in the reading material, and/or vice versa? In the following, we will give

an overview of the relevant research on the interplay of music and text perception and then report an experiment that tests the hypothesis of the mutual emotional impact of music and text.

## Influence of music on cognitive performance

The influence of music on cognitive performance has been extensively investigated. Evidence about the disturbing, enhancing, or neutral nature of this influence, however, is mixed (see [3], for a meta-analysis). Important factors that have been identified in this context are task difficulty [4], music complexity [5], and the personal preference for external stimulation, along with extraversion [6, 7]. As to emotional factors, arousal and mood conveyed by the music have been shown to differentially influence memory performance [8, 9]. In a recent study, Ward et al. [10] found that mood-matching music can improve recall in older adults. However, there is still no consensus about the distracting or supporting impact of music on cognitive performance, and a closer look at the literature reveals a number of gaps and shortcomings. We want to address the issue with the aim of closing one of these gaps by specifically concentrating on the influence of music on emotion perception while reading narrative texts.

## Influence of background music on the reading process

Several studies have examined the influence of music on cognitive aspects of the reading process, such as reading speed, fluency, and comprehension. Findings, however, are inconsistent. For instance, music has mostly been found to distract the reader, but it can also promote concentration (see [4, 11], for reviews). Several studies demonstrate that text comprehension is impaired by background music or noise when compared to reading in silence [12, 13]. Employing an eye-tracking approach, Zhang et al. [14] confirmed this disrupting effect, showing that background music is associated with more re-reading of the text. However, not every music piece has the same impact on the reading process. Thompson et al. [15] discovered that loud and fast background music led to reduced text comprehension, while soft and slow music did not make a difference in this regard. Moreover, the distracting effect has been shown to depend on whether or not the music was preferred as study music [16]. Participants scored significantly lower in a reading comprehension test when they had listened to non-preferred study music while reading than when they had read in silence. Music previously categorized as preferred study music did not have this effect. As to the differentiation of music types, Vasilev et al. [17] concluded in their meta-analysis that background noise, music, and speech have a stronger distracting effect on reading performance when background speech contains meaningful content and when music has meaningful lyrics. However, their analysis found small but stable adverse effects for all kinds of background sounds, which was true for adults as well as for children. Besides the influence of tempo, loudness, preference, type, and other characteristics of the music, interindividual differences have also been shown to affect the strength of the music's impact on reading performance. As such, extraverts and persons with higher working memory capacity seem to be less disturbed by background music than introverts and those with lower working memory capacity [18, 19].

## Emotion perception in music and text

In contrast to the relatively well-studied influence of music and noise on the reading process, there are no reports in the literature on whether music also changes the emotional evaluation of the reading content, and vice versa. We therefore merely provide a brief overview of studies addressing related research questions.

Gfeller et al. [20] investigated the affective response to music, text, or the combination of both. In their design, both text and music stimuli were presented acoustically, as they employed a recorded recitation of a poem and recordings of commercial-style or atonal music. Their results show that some emotional ratings changed when music was combined with text, as compared to music only or text only. For instance, the text was liked better when combined with popular commercial-style music than in combination with unpopular atonal music. The authors conclude that listening to music can influence our emotional response to simultaneously received texts.

Another study on emotional factors in the context of reading with background music by Kallinen [21] showed that valence assessment of texts depended on music tempo. Focusing on gender effects, they found that men rated news texts to be more positive when listening to fast rather than slow background music, whereas women's ratings were most positive in the no-music condition. Unfortunately, the researchers employed only one music piece with different tempo modifications instead of different music stimuli, so it is difficult to generalize their results.

Turning the tables, some attempts have been made to explore the inverse emotional impact of text in the form of lyrics on music perception [22, 23]. In an early work, Stratton and Zalanowski [24] revealed a relative dominance of emotions conveyed by the lyrics compared to emotions carried by the music. More recently, Ali and Peynircioğlu [25] demonstrated that sad music with sad lyrics is perceived as sadder than sad music without lyrics. Interestingly, happy music with happy lyrics was rated as less happy compared to happy music without lyrics. The authors concluded that congruent lyrics detract from and thus attenuate happy emotions, while they bolster sad emotions conveyed by the music. In contrast to the findings by Stratton and Zalanowski [24], Ali and Peynircioğlu found that the emotions carried by the music are dominant when music and lyrics convey incongruent emotions (see also [26]). In agreement with the latter results, Brattico et al. [27] showed that happy music without lyrics activates the limbic system whereas happy music with lyrics is reflected in auditory regions only. Except for the described research on music and lyrics, an examination of the influence of nonmusical text on music perception has not yet been made.

### Influence of expertise on multisensory perception

In multisensory research, it is of great interest to examine differences between experts and novices. For instance, it has been shown that persons who scored higher on a test that assessed their touching expertise by measuring 'need for touch' were less susceptible to the nondiagnostic haptic experience of a container when rating the gustatory qualities of its contents [28]. In the context of music and reading, there are mixed results regarding the impact of musical expertise on cognitive tasks. Patston and Tippett [29] found that musicians but not non-musicians were disturbed by background music in the completion of a language comprehension task (see also [30]). However, musicians benefit more from soft background music in terms of attention performance than do non-musicians [31]. As there is no clear evidence about musical expertise as a factor in the susceptibility to background music, we want to address this issue in the present research.

### Emotional mediation of multisensory processes

In the literature on multisensory associations or crossmodal correspondences [32], it is widely acknowledged that the emotional state a person experiences during the reception of the multisensory input plays an important role. When stimuli from one sensual modality influence the perception in another, or when multisensory stimuli are consistently associated with each

other, emotional states have often been found to explain the effects [33, 34]. The emotion mediation hypothesis [35] suggests that stimuli which trigger the same emotion when presented separately, are more likely to be perceived as fitting together (see [36], for a review). This connection was demonstrated for associations among, inter alia, odor and color [37], music and color [38], music and taste [39], taste and shape [40], and touch and color [41].

As we want to examine the potential influence of music mood on perceived text mood and vice versa, we must separately consider the emotions attributed to the stimuli on the one hand, and the recipient's mood state on the other hand. Only if both affective dimensions are independently measured, can emotional mediation be examined. In our study, the stimulus attributes music mood and text mood were assessed in a pre-test with regard to the extent to which they conveyed a happy or a sad mood. The respective texts and music pieces were then categorized as 'happy' or 'sad'. The dependent variables 'perceived text mood' and 'perceived music mood' were measured by numeric rating scales. We employed a pictorial scale to determine the mood state of the participant. According to the emotion mediation hypothesis, the independent mood variables should lead to a change in experienced emotions, which in turn would correlate with the dependent mood measure. That is, sad music would make the participant feel sadder, which in turn would lead to higher sadness-ratings of the text, and vice versa.

### Inducing emotions through music and texts

Numerous studies have shown that music reliably evokes emotions in the listener [1, 42]. This has proven very useful when it comes to the focused elicitation of specific emotions as an experimental method [43, 44]. Especially for happiness and sadness, playing music to participants turned out to be a very effective induction method [45]. Likewise, narrative texts have been used to induce emotions in experimental studies [2], as speech per se is known to easily convey mood, be it via single words, sentences, or whole stories [46–48].

### Emotions influence music and text perception

Research on the impact of mood on emotion perception in texts or music has received only limited attention so far. As to music, Vuoskoski and Eerola [49] found that participants' current mood is associated with mood-congruent music ratings. However, this effect was moderated by extraversion, suggesting a complex relationship between mood and personality. Another study reported distinct age differences in the influence of listener mood on affect perception in music. Younger adults' perception of negative valence in the music was mood-congruent, whereas it was mood-incongruent for older adults. Arousal perception in the music was mood-congruent for both age groups, but the effect was much more pronounced in younger participants [50]. In the context of text perception, the prevailing mood has thus far only been considered in terms of reading performance. Mills et al. [51] showed that a sad mood leads to more profound text comprehension (deep-reasoning) compared to a happy mood (see also [52, 53]). Opposite results were found by Scrimin and Mason [54], whose participants showed better text comprehension and better learning of factual knowledge when they were in a positive mood. Apparently, evidence about the connection between the receiver's emotions and music and text perception is still limited. Exploring this relationship will be part of this study.

### Our study

The main objective of this work is to investigate the relationship between perceived emotions in music and text when simultaneously consumed. We want to examine if a distinct mood in the music is reflected in the mood evaluation of a text being read at the same time, and vice

versa. We are also interested in exploring a potential influence of emotions conveyed by music or text on the perceived music-text fit, as well as on the receivers' felt immersion into the music or the text. As we are interested in both directions of the potential crossmodal effects, we applied a symmetrical research design. A secondary aim of the study is to regard the impact of relative expertise on multisensory transfer and association. We furthermore examine the assumption of an emotional mediation for all mentioned effects.

We hypothesized that the emotional mood of the music (or the text) influences the perceived mood conveyed by the text (or the music) by way of an emotional transfer. We assumed that this effect is less pronounced in experts than in novices.

To test these hypotheses, we performed two pre-tests and one laboratory experiment with three separate samples. The pre-tests validated the emotional value of music pieces and narrative text excerpts and were the basis to choose the stimuli for the main experiment. In a fully crossed within-subjects-design, participants were asked to read the texts while listening to the music and to rate both text and music in terms of the conveyed emotional mood (happy vs. sad), artistic quality, perceived immersion, and general liking. We also assessed participants' momentary emotional state by measuring valence, arousal, and dominance at the beginning of the experimental session and after every trial.

## Method

### Sample

40 students (31 female, 9 male; $M_{age}$ = 23.95, $SD$ = 4.27) volunteered to participate in the experiment, amongst which 39 were psychology students and one studied another subject. As it is mandatory for psychology students at the Johannes Gutenberg University Mainz to participate in studies conducted in the department, most of them had prior experience with cognitive experiments. However, there was no overlap or potential interference with designs tested here. All of them gave written informed consent in accordance with the Declaration of Helsinki. 38 participants had self-reported normal or corrected-to-normal vision and hearing; only two were slightly myopic and did not wear glasses or contact lenses. However, their data did not deviate substantially from those of the rest of the sample, so they were included in the subsequent analyses. All participants reported being non-synesthetes. As all subjects were university students in good standing, we assumed that their cognitive performance was not impaired. Thus, we did not carry out cognitive tests, nor did we test them for previous substance use. Musical expertise was assessed by the Goldsmith Musical Sophistication Index Version 1, a self-report inventory for individual differences in musical sophistication (Gold-MSI; [55]). It measures the ability to engage with music independently of musical style, preferences, or regularity of musical practice. In order to avoid overloading the questionnaire, we picked 18-items assessing the General Factor of Musical Sophistication. In our sample, the mean General Musical Sophistication Score (GMS) was below average compared to the norm ($M_{GMS}$ = 68.97, $SD_{GMS}$ = 23.13; 28th percentile; norm: $M_{GMS}$ = 81.58, $SD_{GMS}$ = 20.62). One participant scored extremely high ($GMS$ = 122); however, her data did not stand out and were thus not dismissed. For analysis, we performed a median split to divide the sample into more (experts) and less (novices) musically experienced participants (median $_{GMS}$ = 69.25). We identified extreme outlier values in terms of music and text ratings with the help of boxplots (criterion: > 3 * IQR from 1st or 3rd quartile) and replaced these outliers by the group mean. In total, this was the case for five data points representing 0.21% of all ratings. In addition to the Gold-MSI questionnaire, we had asked for daily music listening and reading routines. Most of the sample reported listening to music (almost) every day ($n$ = 35), the remaining participants indicated to do so less frequently. In terms of reading, 15 participants reported to read (almost) every

day, 10 read several times a week, 13 read several times a month and two participants indicated to (almost) never read for pleasure. 6 participants found reading (more or less) tiring, and one participant indicated only to read when necessary. Most of the sample ($n = 33$) reported that they did (rather) not like to listen to music while reading.

As the group of those used to simultaneous reading and listening to music was very small ($n = 7$), we could only exploratively examine if habituation might be an influencing factor for the crosstalk between music and text. The experiment was conducted in accordance with the Declaration of Helsinki. Informed consent was obtained before the experimental session, subjects were told that they could abort the experiment at any time without consequences, and they were debriefed about the purposes of the study immediately after the session. The study was approved by a local Departmental Ethics committee under a blanket approval.

## Apparatus and stimuli

**Setup.** The experiment took place in a darkened laboratory room equipped with a table and an office chair. The table was standing in front of a wall of LED panels with movable walls to the left and right of the table so that the walls constituted an open booth (see Fig 1). The LED panels emitted slightly reddish light (RGB: 255, 55, 0), creating a comfortable, warm atmosphere. An additional desk lamp on the table provided customary reading light. Noise-canceling headphones lay ready on the table.

**Hard- and software.** Stimuli were generated with a Dell Optiplex 980 PC (Core i5) using Python 2.7.13. Data were analyzed with the statistics software IBM SPSS statistics version 27.

**Text stimuli.** We employed excerpts from four different belletristic texts that have been shown to convey either sad or happy emotions, drawing on Zupan's and Babbage's list of

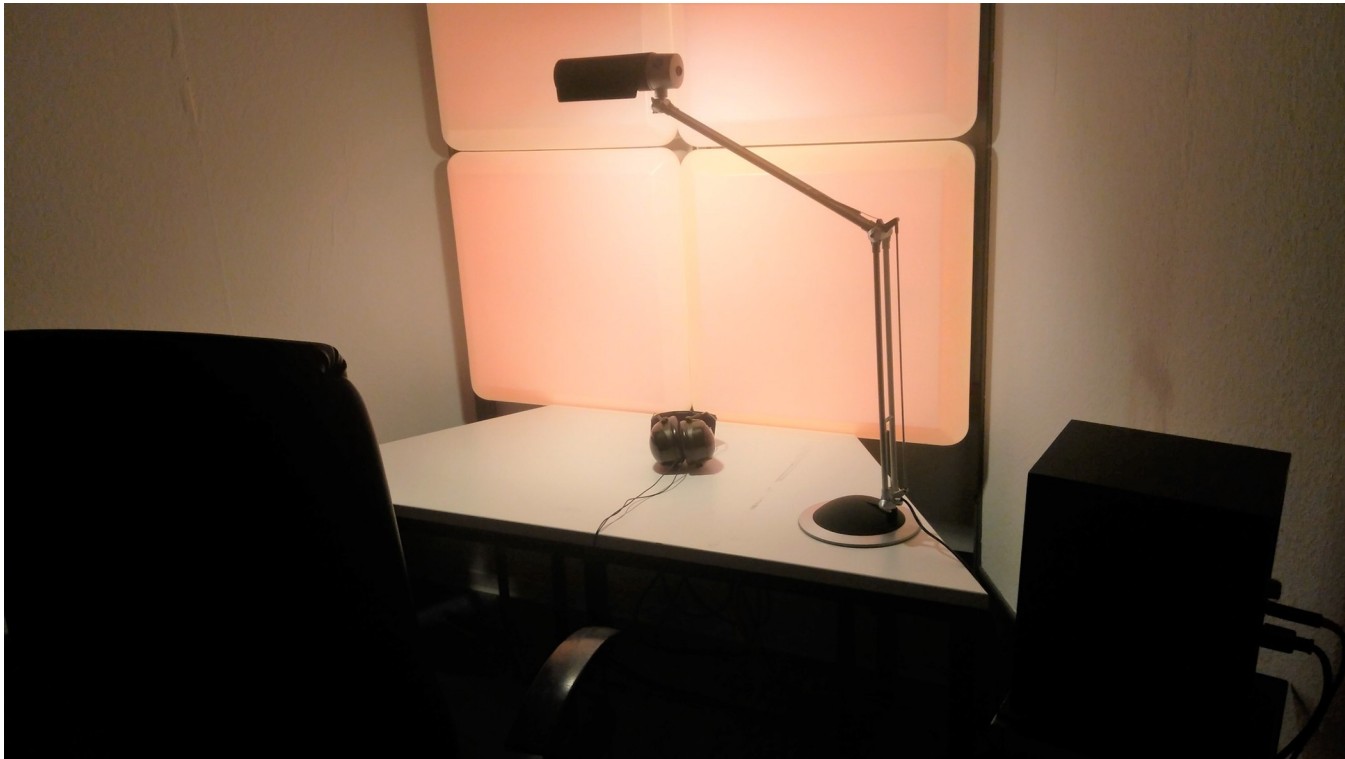

**Fig 1. The experimental setup.** The LED wall emitted a slightly reddish light creating a comfortable atmosphere, a table lamp facilitated reading in the otherwise dark room.

narrative texts for emotion elicitation [2]. In a small pilot experiment with an independent sample ($n$ = 8, 6 female; $M_{age}$ = 31.38, $SD_{age}$ = 9.47), we tested the emotional value and impact of six happy, six sad, and six neutral excerpts from novels that had been translated into German. The texts had a length of 500 to 600 words and required approximately three minutes of reading time on average. Participants rated the emotional mood of the text on a bipolar 11-point scale from -5 (sad) to +5 (happy) and the intensity of the mood on a 10-point scale from 0 ('not at all') to 9 ('very much'). As there is no generally accepted tradition of mood assessment for crossmodal stimuli, we have opted for an integration of qualitative and quantitative measurements. For the analysis, we generated a mood score by multiplying the two ratings. Participants furthermore rated valence, arousal, and dominance they felt at that moment on a 9-point version of the Self-Assessment Manikin scales (SAM scales [56]). We chose the two texts each that had been rated as most happiness- and sadness-conveying as stimuli for the main experiment. For the experimental sessions, the texts were printed on custom white print paper (size Din A 4). Details about the chosen stimuli–two sad and two happy text excerpts– can be retrieved from S1 Table.

**Audio stimuli.** In analogy to the text stimuli, we chose four music pieces as audio stimuli said to carry either happy or sad emotions. We pretested six happy and six sad pieces that had been used to induce emotions in previous studies (see [57], for a review). All pieces were classical string music from the 18th or 19th century, which we cut or looped to a length of four minutes each. To reassure that the musical character would not vary too much throughout each piece, we cut the audio track just before a distinct mood change, e.g., a change of key, and restarted the piece from the beginning, until four minutes were reached. In an online-pretest, 45 participants (33 women; $M_{age}$ = 25.64, $SD_{age}$ = 7.15; $M_{GMS}$ = 65.68, $SD_{GMS}$ = 23.7) rated the pieces on the same scales as had been used for pretesting the text stimuli. We chose the music pieces that scored highest in terms of conveying a happy and a sad mood (two music pieces per mood) for the main experiment. S2 Table lists further information about the chosen stimuli. The resulting mp3-files were set to an equivalent continuous sound level of about 60 dB using an SPL-meter (NTi AL1). During the experimental session, they were presented through headphones.

## Design and procedure

We applied a fully crossed 2 (music mood) x 2 (text mood) within-subjects design, resulting in four experimental conditions. To avoid the repetition of a text or music piece for any given subject, there were two texts and two music pieces per mood (two happy, two sad each). In order to counterbalance potential order effects, all possible stimulus orders were presented to the same number of subjects. The four combinations per condition resulted in eight possible permutations of the four conditions, each presented to 5 of the 40 participants. Their assignment to a given stimulus order was randomized.

Participants were tested individually. First, their prevailing mood was assessed through the SAM scales (see stimuli section), then they were asked to put on the headphones and read the first text. The music started as soon as they began reading. Participants were told to focus on the text, just as if they were reading for pleasure in their leisure time. They were instructed to continue reading for as long as the music would play, and start from the beginning of the text in case they would have finished reading it before the end of the music. With this instruction, we ensured that every participant read the full text and heard the entire music piece. When the music stopped, participants filled in the SAM scales again, followed by a questionnaire measuring their liking of the text ('How do you like the text?'), their evaluation of its artistic quality ('How would you rate the artistic level of the text?'), their perceived immersion ('How much

did the text intrigue you?'), and their assessment of the text's emotional mood ('How would you rate the emotional mood of the text?'). Afterward, the same set of questions was filled in to evaluate the music, supplemented by an indication of familiarity with the piece ('How well did you know the music before this experiment?'), and followed by a question on the perceived fit of text and music ('How would you personally rate the fit between the text and the music?'). All constructs, except for emotional mood, were measured on 10-point scales from 0 ('not at all') to 9 ('very much'). Like in the pretests, the character of the emotional mood was rated on a bipolar 11-point scale from -5 (sad) to +5 (happy), followed by the measurement of emotional intensity on the noted 10-point scale. Half of the participants first rated the texts and then the music, while the other half received the questionnaires in reverse order. Reading speed was measured with a stopwatch. The whole sequence was repeated for all four experimental conditions.

At the end of the experiment, participants filled in a final questionnaire (see Supplementary Material) assessing demographic data as well as different control variables, namely eyesight, hearing, synesthesia, gender, age, profession, highest completed level of education, reading habits, music listening habits, reasons for listening to music, activities combined with listening to music, attitudes towards reading, and the short version of the Gold-MSI [55] measuring musical expertise. The experimental session ended after approx. 35 minutes. The temporal sequence of the experimental session can be seen in Fig 2. Data were collected between July 29th 2020 and January 17th 2021.

## Results

In the first step, we will describe analyses on the influence of music on text perception, and in the following, we will consider the reverse influence of text on music perception. In addition to the included graphs and tables, boxplots further illustrating participants' ratings are shown in S1 Fig.

### Influence of music mood on text perception

We calculated a 2 (music category) x 2 (text category) repeated measures multivariate analysis of variance (rmMANOVA) for the dependent variables perceived mood score, quality, immersion, and liking of the text. The factors 'music category' and 'text category' refer to the classification as 'sad' or 'happy' according to the pre-experiments (see methods section). Both multivariate main effects and the multivariate interaction were significant (see Table 1 for the complete results). Post-hoc univariate rmANOVAS (LSD-corrected) showed that the main effect of text category was significant for perceived text-mood score, immersion, and liking, but not for quality. As expected, happy texts were perceived as conveying a happier mood, they were liked better, but they were less immersing than sad texts. The main effect of music category was significant for perceived mood score, that is, texts were rated as conveying a happier mood with happy background music than with sad background music. There were no

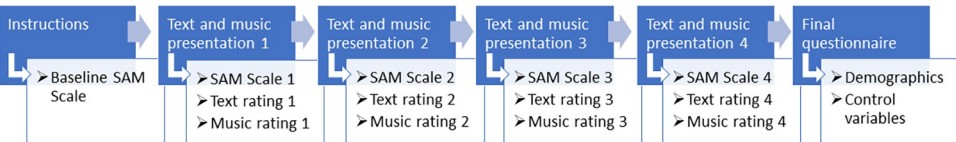

**Fig 2. Temporal sequence of the components of one experimental session.** Note that the flow chart is exemplary for 50% of the participants that were instructed to rate first text then music. The other 50% of the participants started with music ratings. Participants were randomly assigned to the 'text first' or 'music first' groups.

**Table 1. Results of a 2 (music category) x 2 (text category) rmMANOVA on perceived mood score, quality, immersion, and liking of the text.** Multivariate and univariate main effects and interactions.

| Music or text dimensions | | $F$ | df | $p$ | $\eta^2$ |
|---|---|---|---|---|---|
| Multivariates[a] | Univariates | | | | |
| **Music category** | | **2.89**[*] | **4 / 36** | **.036** | **.243** |
| | Text-mood score | 12.34[*] | 1 / 39 | .001 | .240 |
| | Text quality | 1.37 | 1 / 39 | .249 | .034 |
| | Text immersion | 1.69 | 1 / 39 | .201 | .041 |
| | Text liking | 1.22 | 1 / 39 | .276 | .030 |
| **Text category** | | **70.43**[**] | **4 / 36** | **< .001** | **.887** |
| | Text-mood score | 259.27[**] | 1 / 39 | < .001 | .869 |
| | Text quality | 1.11 | 1 / 39 | .299 | .028 |
| | Text immersion | 17.90[**] | 1 / 39 | < .001 | .315 |
| | Text liking | 4.56[*] | 1 / 39 | .039 | .105 |
| **Music category x text category** | | **3.72**[*] | **4 / 36** | **.012** | **.292** |
| | Text-mood score | 2.47 | 1 / 39 | .126 | .059 |
| | Text quality | 6.05[*] | 1 / 39 | .018 | .134 |
| | Text immersion | 11.31[**] | 1 / 39 | .002 | .225 |
| | Text liking | 4.36[*] | 1 / 39 | .043 | .101 |

[a]Multivariates refer to Pillai's trace. Asterisks indicate significant effects ([*]: $p < .05$; [**]: $p < .01$).

significant univariate effects of music category for text quality, immersion, and liking. Univariate interactions of music category x text category were significant for text quality, immersion, and liking, but not for text-mood score. As illustrated in Fig 3, texts were perceived as higher in quality, as more immersing, and they were liked better when read under congruent music. All effects were clearly more pronounced for sad compared to happy texts.

We then tested if the found effects of music mood on text evaluation were mediated by the experienced emotions of the participant. As mentioned above, experienced emotions were measured according to their valence, dominance, and arousal at the beginning of the experiment (baseline) and after every trial. For analysis, we subtracted all values from the baseline measurement and hence displayed the relative change in the emotional state throughout the experimental session. Then, we first calculated a 2 (music category) x 2 (text category) rmMANOVA on valence, arousal, and dominance, and, second, examined the interrelation of the significant emotion measures with the different text evaluation variables by calculating Pearson correlations.

The rmMANOVA revealed a significant main effect for music category, thus the stimulus mood did influence the participants' emotional condition, $F(3, 36) = 4.5$, $p = .009$, $\eta^2 = .27$. Post-hoc univariate rmANOVAs showed that the effect was driven by changes in valence and dominance ratings, while arousal was not significantly affected (valence: $F(1, 38) = 11.25$, $p = .002$, $\eta^2 = 0.228$; dominance: $F(1, 38) = 5.74$, $p = .022$, $\eta^2 = .131$; arousal: $F(1, 38) = 0.85$, $p = .364$, $\eta^2 = 0.022$). In the next step, however, valence and dominance did not reveal any relation to text evaluation: Among all Pearson correlations between valence and perceived text mood, only one out of four possible music-mood-text-mood-combinations was significant (sad music with happy texts, $r(38) = .32$, $p = .042$). In addition, for none of the four combinations did we find a significant correlation between dominance and text mood, nor between valence or dominance and liking of the text. Nevertheless, text immersion correlated positively with valence for happy-music-happy-text-combinations, and negatively for happy-music-sad-text-,

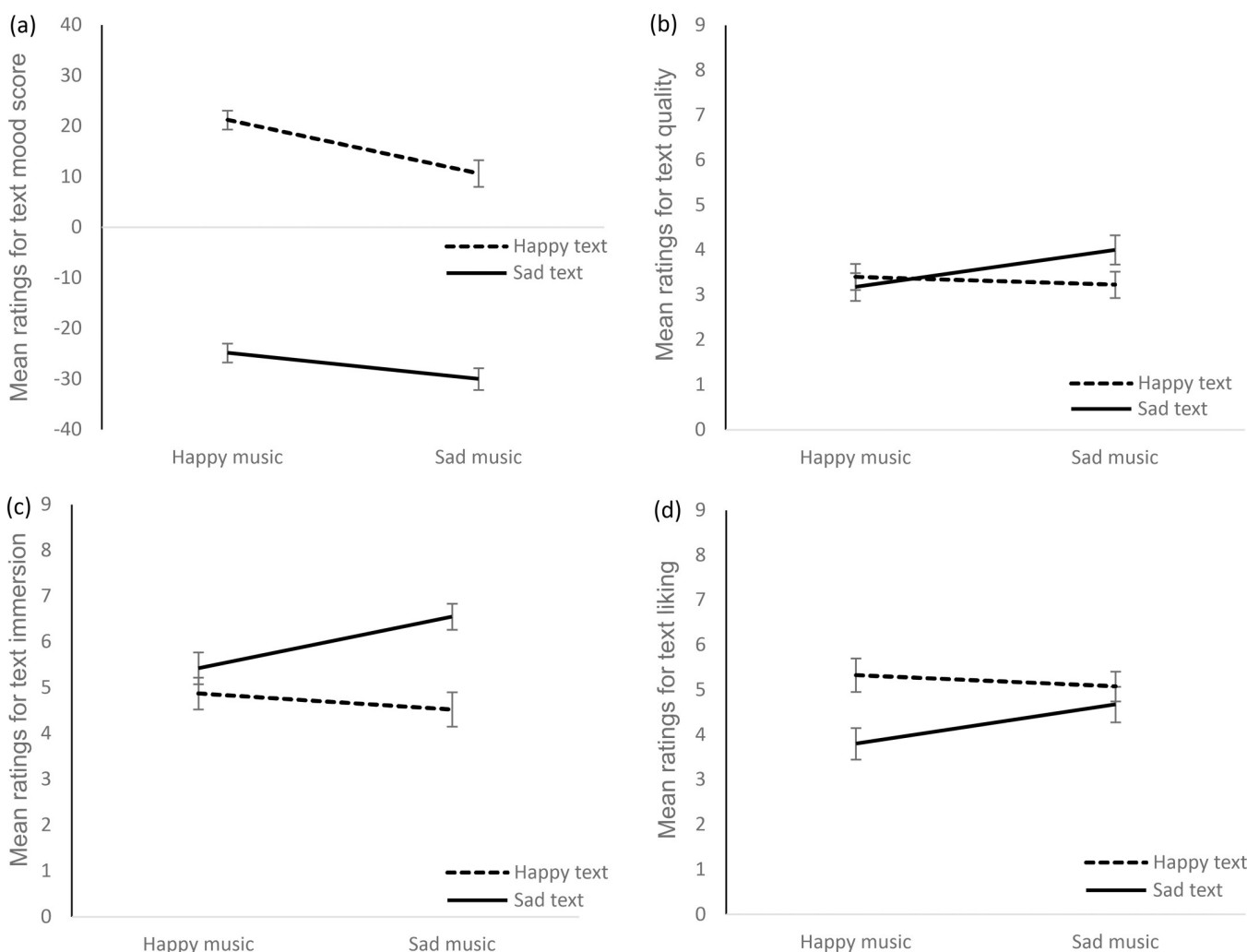

**Fig 3.** Means for perceived mood score (a), artistic quality (b), immersion (c), and liking (d) of the text as a function of happy vs. sad music and text categories. Error bars show ± 1 SEM.

and sad-music-sad-text-combinations (see S3 Table, for all correlation values). Data indicated that valence mediates the effect of music mood on text immersion.

## Influence of text mood on music perception

To examine the reverse influence of text mood on music perception, we calculated a 2 (music category) x 2 (text category) rmMANOVA for the dependent variables perceived mood score, quality, immersion, and liking of the music. Both main effects of music category and text category were significant, as well as the respective interaction (see Table 2, for the complete results). Univariate rmANOVAS showed that text category significantly influenced the perceived mood score of the music. Music-mood was rated as happier when participants were reading a happy text while listening. The univariate interaction of music category x text category was significant for liking and immersion of the music. Music was liked better while reading a congruent compared to an incongruent text. Similarly, music was rated as more immersing when presented simultaneously with a congruent rather than an incongruent text. As illustrated in Fig 4, this effect was stronger for sad than for happy music.

**Table 2. Results of a 2 (music category) x 2 (text category) rmMANOVA on perceived mood score, quality, immersion, and liking of the music.** Multivariate and univariate main effects and interactions.

| Music or text dimensions | | $F$ | df | $p$ | $\eta^2$ |
|---|---|---|---|---|---|
| Multivariates[a] | Univariates | | | | |
| **Music category** | | **56.22**** | **4 / 35** | **< .001** | **.865** |
| | Music-mood score | 215.3* | 1 / 38 | < .001 | .850 |
| | Music quality | 24.44** | 1 / 38 | < .001 | .391 |
| | Music immersion | 0.00 | 1 / 38 | .977 | .000 |
| | Music liking | 10.96* | 1 / 38 | .002 | .224 |
| **Text category** | | **5.64*** | **4 / 35** | **.001** | **.392** |
| | Music-mood score | 18.55** | 1 / 38 | < .001 | .328 |
| | Music quality | 0.03 | 1 / 38 | .860 | .001 |
| | Music immersion | 1.48 | 1 / 38 | .231 | .038 |
| | Music liking | 0.38 | 1 / 38 | .543 | .010 |
| **Music category x text category** | | **3.59*** | **4 / 35** | **.015** | **.291** |
| | Music-mood score | 3.89 | 1 / 38 | .056 | .093 |
| | Music quality | 0.02 | 1 / 38 | .881 | .001 |
| | Music immersion | 6.01* | 1 / 38 | .019 | .137 |
| | Music liking | 6.07* | 1 / 38 | .018 | .138 |

[a]Multivariates refer to Pillai's trace. Asterisks indicate significant effects (*: $p < .05$; **: $p < .01$).

To test the emotional mediation of the influence of text mood on music perception, we mirrored the analytical procedure described above for the reverse effect. The rmMANOVA on emotional measures revealed a main effect of text category, $F(3, 36) = 14.06$, $p < .001$, $\eta^2 = .053$, which showed significance in all three univariate analyses, indicating that the stimulus mood had an influence on the participants' emotional state (valence: $F(1, 38) = 33.78$, $p < .001$, $\eta^2 = .471$; arousal: $F(1, 38) = 11.23$, $p = .002$, $\eta^2 = .228$; dominance: $F(1, 38) = 29.28$, $p < .001$, $\eta^2 = .435$). Sad texts were associated with lower valence, but higher arousal and dominance ratings compared to happy texts. However, there were no significant correlations of valence, arousal, and dominance ratings with perceived music-mood score for none of the four music-text-mood-combinations. For sad-music-sad-text-combinations, there was a significant positive correlation between arousal and music liking and a significant negative correlation between valence and music immersion (see S4 Table, for all correlation values). A general mediation by emotion ratings could not be observed.

## Music-text fit

To test if music and text are more likely to be perceived to fit together when they convey the same mood, we pursued two alternative approaches. First, we inspected the interrelation between our stimulus mood categories (happy vs. sad) and music-text fit ratings. Therefore, we calculated a 2 (music category) x 2 (text category) rmANOVA with music-text fit as the dependent variable, and found a strong music category x text category interaction, $F(1, 39) = 111.55$, $p < .001$, $\eta^2 = .741$. Fit ratings were higher when music and text categories corresponded. Complete results can be retrieved from S5 Table.

In a second approach, we took participants' mood ratings as a reference and determined the absolute difference between music-mood ratings and text-mood ratings for each combination of music and text categories. We then calculated Pearson correlations between this difference value and fit ratings. All four correlations were negative, with significant negative

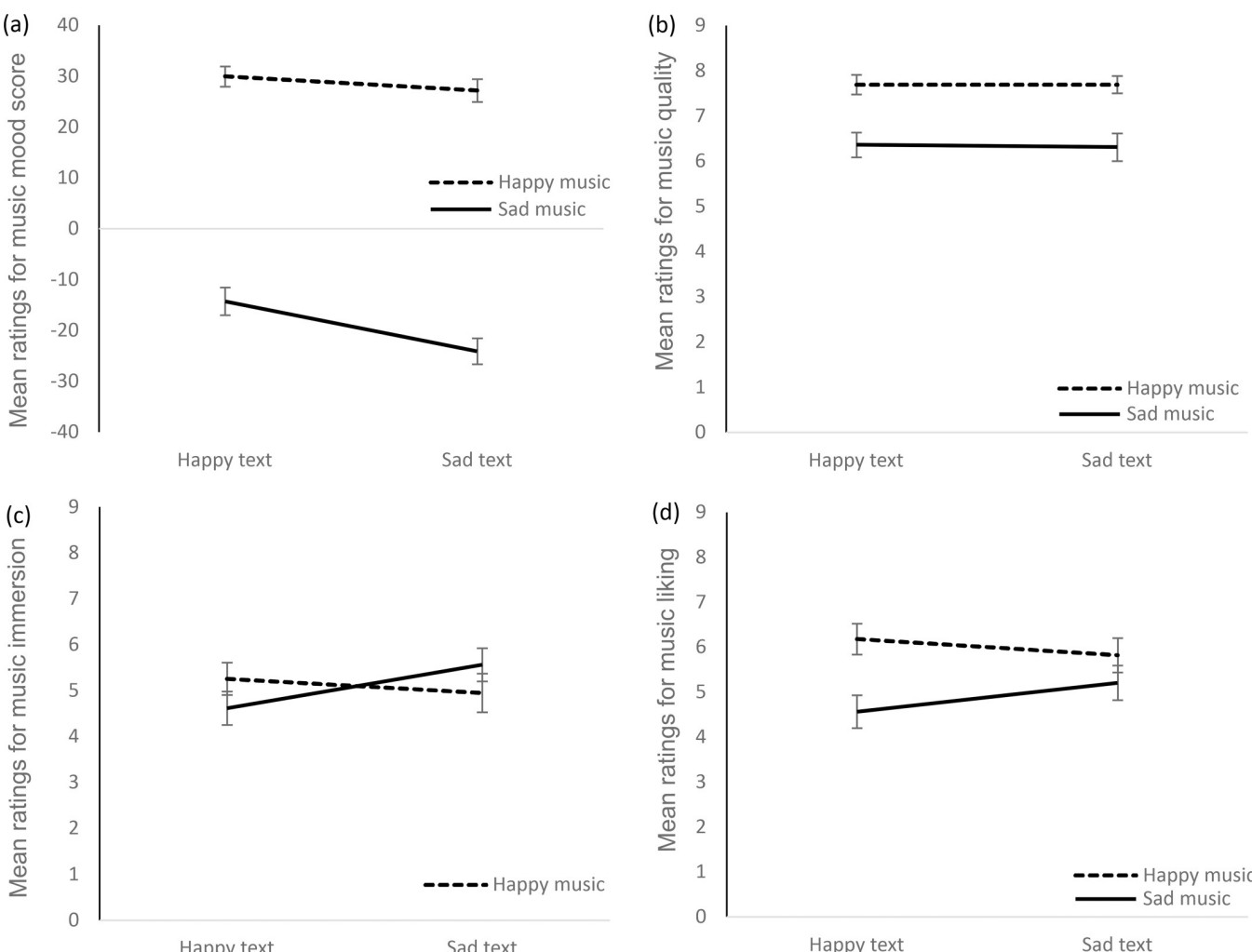

**Fig 4.** Means for perceived mood score (a), artistic quality (b), immersion (c), and liking (d) of the music as a function of happy vs. sad music and text categories. Error bars show ± 1 SEM.

correlations for happy-music-sad-text combinations, $r(38) = -.33$, $p = .039$, and for sad-music-happy-text combinations, $r(38) = -.56$, $p < .001$. The more similar the respective moods conveyed by text and music were rated, that is, the smaller the perceived mood difference, the better music and text were perceived to fit together.

## Exploration of the influence of habituation and expertise on the crosstalk between music and text

In our post-questionnaire, we collected data on musical and reading expertise, as well as music listening and reading habits. We used these data to explore if expertise and habituation had an impact on the found crossmodal effects.

**Musical and reading expertise.** To test the influence of musical expertise on the effect of music mood on text perception, we calculated the same 2 (music category) x 2 (text category) rmMANOVA for perceived mood score, quality, immersion, and liking of the text, this time with the between-subjects-factor musical expertise. This factor has been built by performing a median split on the Gold-MSI Musical Sophistication Score (see methods section) to receive

the two groups of musical experts and non-experts. There was no significant effect of musical expertise for any of the dependent variables (perceived mood score: $F(1, 38) = 0.37$, $p = .547$, $\eta^2 = .01$; quality: $F(1, 38) = 0.003$, $p = .995$, $\eta^2 = .0$; immersion: $F(1, 38) = 0.19$, $p = .666$, $\eta^2 = .005$; liking: $F(1,38) = 0.001$, $p = .976$, $\eta^2 = .0$), nor an interaction with one of the main factors (see S6 Table). We calculated another 2 (music category) x 2 (text category) rmMANOVA on perceived mood score, quality, immersion, and liking of the music with the between-subjects factor reading frequency. The latter is a dichotomous variable created from the answers to the 4-point question 'How often do you read for pleasure (books, newspapers, magazines, news texts; analog or digital)?' by contrasting frequent readers ('several times a week' and more) with seldom readers ('several times a month' and less). There was a significant between-subjects effect of reading frequency for liking of the music ($F(1, 37) = 5.11$, $p = .030$, $\eta^2 = .121$). Frequent readers gave generally higher ratings for music liking than seldom readers. There were no other significant univariate between-subjects effects (perceived mood score: $F(1, 37) = 0.41$, $p = .526$, $\eta^2 = .011$; quality: $F(1, 37) = 0.43$, $p = .518$, $\eta^2 = .011$; immersion: $F(1, 37) = 1.15$, $p = .291$, $\eta^2 = .030$), nor significant interactions of reading frequency with the main factors (see S7 Table).

**The habit of listening to music while reading.**   We had asked participants to indicate their degree of agreement or disagreement with the statement 'I like to listen to music while reading'. Most of the participants ($n = 33$ out of 40) said (rather) not to do so. Even though the remaining group of 'music-readers' was very small, we were interested in potential differences between the 'music-readers' and the 'silence-readers' in terms of music- and text-mood interplay. Thus, in an explorative approach, we calculated the 2 (music category) x 2 (text category) rmMANOVA for perceived mood score, quality, immersion, and liking of the text again, now with the between-subjects-factor habit. We carried out an analogous rmMANOVA with the same factors and dependent variables referring to music perception. There was a significant effect of habit for text mood, $F(1, 38) = 5.525$, $p = .024$, $\eta^2 = .0127$. As can be seen in Fig 5, text-mood ratings by music-readers were more strongly influenced by the music than those of silence-readers. There were no other significant between-subjects effects for habit, nor an interaction of habit with the main factors (see S8 Table).

## Discussion

The present research examined the interrelation between emotional appraisals of classical music and narrative texts. In a fully crossed within-subjects design, participants were asked to read two sad and two happy texts while listening to sad or happy music. Subsequent ratings referred to the emotions perceived in the stimuli, but also to the participants' current emotional state. We furthermore assessed perceived immersion, as well as artistic quality, liking, and perceived music-text fit. The study revealed three key findings: First, we showed that both music and text have a mutual influence on the emotional appraisal of the other, suggesting a transfer of emotions. Second, it turned out that both music and text are perceived as more immersing and are liked better when music and text mood correspond. Third, we saw that the more similar the separate mood ratings of music and text, the better they were perceived to fit together. Importantly, only the crossmodal effect on text immersion was mediated by valence, whereas all other effects were not mediated by the emotions experienced by the participant. We also examined potential group differences, but neither musical expertise nor reading frequency made a difference in terms of the reported outcomes. However, an explorative examination of reading habits hints at a difference between those who are used to reading while listening to music compared to those who are not combining these two activities.

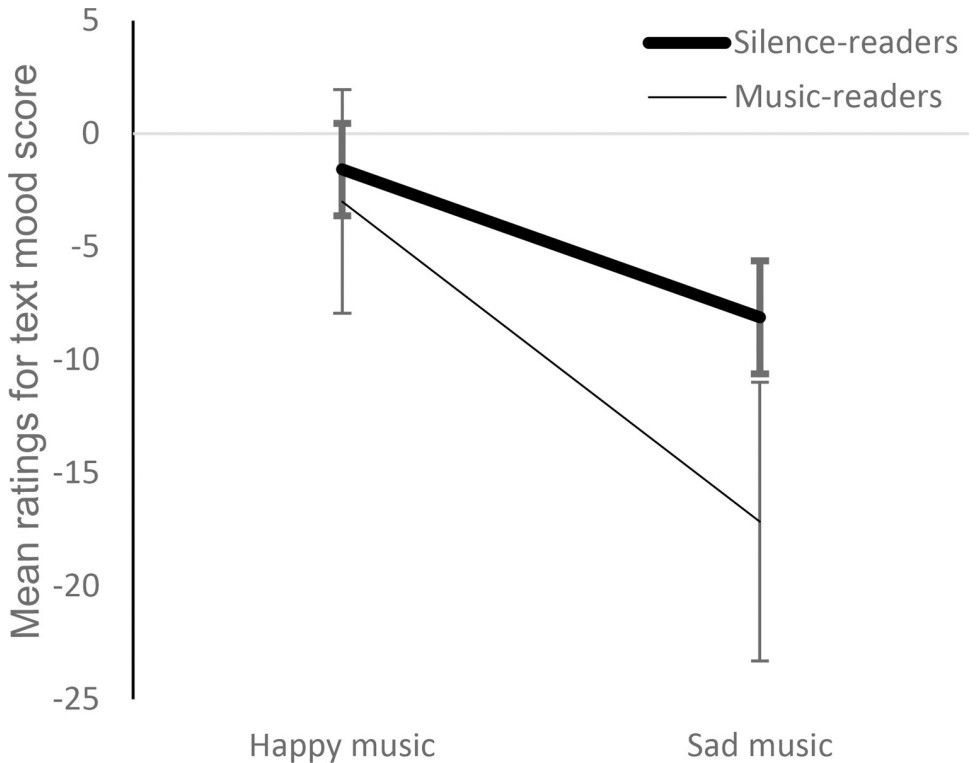

**Fig 5. Mean ratings for text mood score as a function of the habit to listen to music while reading.** Error bars show ± 1 SEM.

## Multisensory transfer of emotional attributions

As expected, our findings indicate that happy or sad mood is transferred from music to texts and vice versa. That is, sad (happy) music makes written stories appear sadder (happier), and sad (happy) stories make music sound sadder (happier). This observation agrees well with numerous studies in the field of multisensory research that have examined the influence of music on the perception of diverse sensual stimuli. For instance, it has been found that listening to 'soft' music increases haptic softness perception [58], listening to 'light' music makes red wine taste lighter [59], music can enhance the emotions conveyed by pictures [60], and coffee tastes less bitter in the presence of 'sweet' music [61]. In a recent study by Ehret et al. [62], musical attributions were also shown to affect the perceived overall valence of the atmosphere. Accordingly, Yamasaki et al. [63] demonstrated that the evaluation of the environment can be shifted in the direction of specific characteristics of the music. Our results integrate these previous outcomes, as they demonstrate the mutual impact of auditory input on visually received information, and vice versa, which then influence the overall impression of a fictive environment and atmosphere.

Previous authors have often solicited emotion as the mediator and key to deciphering the crossmodal transfer of stimulus characteristics (see introduction). Our research disagrees with this interpretation, as emotional mediation did not explain the effects of crossmodal transfer and association in the case of music and text. However, the comparability of study designs is limited. Narrative texts are complex stimuli and the task of rating the conveyed mood of the text implies not only visual processing but also the cognitive effort of reading. In contrast, it is common in the field of multisensory research to employ simple stimuli like color patches,

single tones, and basic taste samples. With our approach, we offer a first assessment of cross-modal emotional mediation involving texts as stimuli.

Regarding the underlying emotions, we can conclude that music and text did not primarily affect the participants' mood but rather did the music impinge on the assessment of the text. And vice versa did the text alter the appraisal of the music. Thus, we can assume that we found rather pure appraisal effects here. We may ask how likely such intersensory appraisal changes are to influence our actions. Negative emotions were recently shown to influence our actions, specifically our inhibitory control [64, 65]. Note, however, that the crossmodal appraisal effect in the current study are once removed from the actor's emotional state. We can thus only speculate if negative emotions identified in a given text or music piece will impact our behavior. Further studies building on this hypothesis would be a promising path forward.

In our paradigm, most effects were more pronounced for sad than for happy stimulus moods. This is once more in line with the insights by Ehret and colleagues [62], who revealed that adding a negative aspect to a positive ambiance has a bigger impact on the overall evaluation of the atmosphere than does add a positive aspect to a negative status quo. This pattern might be explained by a general negativity bias, suggesting that negative information is given more weight than positive information, it is processed faster and commands more attention [66].

Now that the mutual influence of text and music mood is established, which are the underlying processes? We will refer to two approaches to explain the crossmodal effects. First, it is conceivable that multisensory integration takes place, despite the instruction to merely evaluate one modality. Information that is received in one sensory channel may spill over to the other channel. For the recipient, it is difficult if not impossible to distinguish which information is transmitted via which sensory channel. Impressions and characteristics belonging to one stimulus merge with the other, equivalent to a direct transfer of attributions [67, 68]. In our study, the crosstalk of text and music was reciprocal, that is, music had an influence on text perception, and text mood, in turn, affected music perception. This indicates that the mood conveyed by music and text is indeed regarded as one unified experience [69]. Second, Kahneman's capacity model for attention [70] can be used to explain our findings on a broader level. It claims that cognitive capacity is limited and therefore attention has to be allocated among possible mental activities. As attentional resources become insufficient to closely attend to both text and music, the perceptual system may devote the resources to the recognition of emotion at the expense of linking the perceived emotion to its source. Moreover, it was highly probable within the scope of our 2 x 2-design that text and music transmitted a similar emotional message. This coincidence may have been exploited in the face of high attentional demand.

## Influence of mood congruency on perceived quality, immersion, and liking

In our study, music and text were rated higher in immersion and were liked better when they were presented together with mood-congruent text or music. Moreover, the perceived text quality was higher when text and music mood corresponded, while music quality ratings were not influenced by the text. These findings on quality and liking are not surprising, as studies agree that stimuli are evaluated more positively when accompanied by congruent information in another apparently irrelevant sensory modality [71–73]. The fact that perceived music quality (in contrast to perceived text quality) was not systematically affected by congruency could be explained by the idea that it is generally more difficult to evaluate a classical music piece rather than a belletristic text. Those who are not educated in classical music might not hear quality differences at all, or they might not dare to judge it. The small variance of music quality ratings in our data supports this assumption.

Let us now find explanations for the outcome that participants felt more immersed in music and text when the stimuli were congruent in mood. We could show that the effect of happy music on text immersion is mediated by experienced valence. That is, happy music makes us happier, which in turn promotes our feeling of being absorbed in the happy text. The feeling of happiness is possibly connected to a certain light-heartedness and momentary open-mindedness which enables us to better engage in the positive reading content and to dive down into the happy story. In the case of sad texts valence is negatively correlated with immersion, that is, the worse we feel, the more we get involved with sad texts. We conclude that a text is more captivating if it is congruent with our own emotional state. As to the reverse connection, our current findings on emotional mediation concerning the influence of text mood on music immersion as well as on music liking remain inconclusive at the point and require further investigations.

Our results show that not only immersion but also liking and perceived quality were higher in the mood-congruent conditions. Thus, it seems that a text or a music piece intrigues us more if we generally like it and if we consider it of good quality. However, the causality between these three constructs is yet to examine. Thinking laterally, interesting analogies can be found in film music research. Studies revealed that music and sound effects influence emotions experienced by the viewer [74] and enhance immersion and suspense [75]. Film music furthermore affects the semantic evaluation of film scenes [76], it can foster empathy with the characters [77], and the content of film excerpts is remembered more easily when music and pictures convey a congruent mood [78]. Just like in our paradigm, music appears as an influential factor for the involvement in the story.

## The perceived fit between music and text

Our analysis confirmed distinct patterns of perceived music-text fit. The more similar music and text were evaluated in mood, the better the stimuli were regarded to match. This result is in agreement with our recent study showing that music and colored ambient lighting are judged to fit particularly well if they carry similar emotional connotations [79]. In the same vein, numerous investigations have identified emotional attributions as drivers for the mapping of music with various stimuli from other sensual domains (see [36], for a review).

## The influence of expertise and habit

In contrast to previous research (see introduction), we did not discover a difference between experts and novices in terms of their susceptibility to nondiagnostic crossmodal cues. Even though there is no consensus about experts being more or less influenced by background music, it is generally agreed that they are not affected in the same way as novices. We conducted additional explorative analyses to examine if there is a difference between those who regularly listen to music while reading and those who do not. Indeed, the 'music-readers' were more strongly influenced by background music than the 'silence-readers'. This is in contrast with our expectations based on the finding that students are less disturbed by background music if they are used to studying with music [80, 81]. Even though the group of 'music-readers' was very small in our sample, we still consider this outcome as highly relevant and worth investigating in future studies. As mentioned before, the deviation of our results from the literature might be explained by the limited comparability of our design with classic crossmodal paradigms, but also by different practices to distinguish the more from the less experienced participants.

**Limitations and future directions.** The research design of the present study was well-suited to examine how emotions in texts are related to emotions in the music listened to while

reading. However, some methodical limitations should be considered in potential follow-up studies. As the experiment was carried out in a university setting, the sample was rather homogenous in terms of age, education, musical expertise, and reading and music listening behavior. Potentially confounding factors such as psychological and neurological disorders, substance use, and current infections were not assessed, which might be recommended for future samples. Further, a more diverse composition of participants would facilitate the comparison of group differences. In terms of expertise, participants should be specifically recruited as true experts and novices rather than distinguishing them post-hoc by a median split. According to a recent review, music psychologists agree that a musician should be defined by at least six years of musical expertise with at least one hour of practice per week [82]. Only a few of our subjects fell into this category. Beyond expertise and habits, it would be interesting to consider different levels of reading fluency and literacy, other cognitive abilities, as well as personality factors. As to the latter, it has been found that extraverts are less prone to be disturbed by background music than introverts when it comes to fulfilling cognitive tasks (see [19, 83, 84], but also [85, 86]). Moreover, the examination of age differences is promising as it is probable that older adults will differ from younger adults in their way to react to multisensory overload [87, 88]. For practical reasons, we employed a very limited range of stimuli, namely classical string music and American belletristic literature. These were chosen to clearly convey a happy or a sad mood. Future research should seek to diversify the scope of variables to investigate if the results can be generalized to other domains.

On a more technical level, authors interested in conducting studies including musical and reading stimuli, should take into account that different reading speed of participants leads to differing combinations of text passages with musical phrases. In our study, we addressed this issue by cutting and looping relatively uniform music excerpts excluding more important changes in loudness, tempo, instrumentation, or mode. A different approach could be to control the reading speed by presenting single sentences on a screen or using monotonous sound bites rather than music to reduce variance in the auditory domain. However, this was not suitable for our study as we wanted to create a reading and listening situation that is as realistic as possible in a laboratory setting. Further, the question of the auditory delivery is crucial to any crossmodal study design including auditory stimuli. We chose over-ear headphones to approximate everyday reading-while-listening situations and to prevent distraction through external noise. Nevertheless, the influence of music transmitted through loudspeakers and thereby influencing the overall room ambiance would be interesting to examine.

Despite of the described limitations, the present study is of great value and an important addition to the field. Its strengths include its novelty in terms of stimulus combinations, its applicability, and its timely relevance. The design is well-balanced between practical relevance and control of conditions and the results can be regarded as a solid base for further investigations.

## Practical implications

The findings presented in this article are especially interesting for the development of digital multimedia tools that create multimodal experiences. For instance, music streaming services might use the insights of our study and potential follow-ups to create specific playlists going with the mood of a given reading content. In this sense, an app could be set such that it instantly adjusts the music background appropriate for the reading content the user has selected. Moreover, the idea of creating specific background music for fiction books is obvious. As we showed that listening to mood-congruent music fosters text immersion, soundtracks for written love stories, dramas, and thrillers would be a genuine enrichment for passionate

readers. Transforming the reading of a book into an audiovisual experience might even make it more attractive to those who hitherto prefer to spend their leisure time watching TV and video streaming services.

## Conclusion

The emotional appraisal of the mood conveyed by music and text that are presented simultaneously is of crossmodal nature. We show that listening to music while reading a text enhances *immersion* into the text, but also into the music, if both convey the same mood. That is, we feel more immersed in a sad story when we listen to sad music, and correspondingly, sad music is more moving in combination with a sad text. The same crossmodal amplification is true for happy stimuli. Thus, music and texts are *liked* better, and they are perceived to *fit together* when they are mood-congruent. Mood congruency also enhances *emotion perception* in the stimuli. Sad (happy) texts appear sadder (happier) together with sad (happy) background music and vice versa. These results were, however, not mediated by the valence, arousal, or dominance experienced by the participant, which makes this research stand out from the majority of crossmodal studies. Future research should investigate the generalizability of the discovered phenomenon to other text and music genres. Thereafter, the exploration should be moved from the laboratory to the field to confirm its practical value. Application fields for these findings range from the multisensory enhancement of private settings to the creation of highly immersive multimedia experiences.

## Supporting information

**S1 Table. List of novel excerpts presented as text stimuli derived from Zupan and Babbage [2].**
(DOCX)

**S2 Table. List of music pieces presented as audio stimuli [44, 89–98].**
(DOCX)

**S3 Table. Pearson's correlation coefficients of valence, arousal, and dominance experienced by the participants with perceived text-mood score, text liking, and text immersion.**
(DOCX)

**S4 Table. Pearson's correlation coefficients of valence, arousal, and dominance experienced by the subjects with perceived music-mood score, music liking, and music immersion.**
(DOCX)

**S5 Table. Results of a 2 (music category) x 2 (text category) rmANOVA on music-text fit.**
(DOCX)

**S6 Table. Multivariate effects and interactions of a 2 (music category) x 2 (text category) rmMANOVA with the between-subjects factor musical expertise on perceived mood score, quality, immersion, and liking of the text.**
(DOCX)

**S7 Table. Multivariate effects and interactions of a 2 (music category) x 2 (text category) rmMANOVA with the between-subjects factor reading frequency on perceived mood score, quality, immersion, and liking of the music.**
(DOCX)

**S8 Table. Multivariate effects and interactions of a 2 (music category) x 2 (text category) rmMANOVA with the between-subjects factor 'habit to listen to music while reading' on perceived mood score, quality, immersion, and liking of the text.**
(DOCX)

**S1 Questionnaire. EM–music first.** To evaluate music and text for the group "music first".
(PDF)

**S2 Questionnaire. ET–text first.** To evaluate music and text for the group "text first".
(PDF)

**S3 Questionnaire. Post-questionnaire.** To assess music listening and reading behavior and demographic data.
(PDF)

**S4 Questionnaire. Pre-questionnaire.** To assess the prevailing mood before the confrontation with any stimuli as a baseline.
(PDF)

**S1 Data. Data of the experiment.** Dataset before outlier correction.
(SAV)

**S1 Fig.** Boxplots to illustrate participants' ratings of dependent variables concerning a) music and b) text stimuli.
(TIF)

**S1 File. Syntax containing all analyses.**
(SPS)

**S2 File. Syntax containing the computation of new variables.**
(SPS)

## Acknowledgments

We thank Agnes Münch for technical support, Ariane Wilhelm and Paavo Käseberg for assisting in recruitment and experimental testing, and Daniel Auras for his help with data entry. The second author (HH) acknowledges the support of the Wissenschaftskolleg zu Berlin (https://www.wiko-berlin.de).

## Author Contributions

**Conceptualization:** Pia Hauck, Heiko Hecht.

**Data curation:** Pia Hauck.

**Formal analysis:** Pia Hauck.

**Investigation:** Pia Hauck.

**Methodology:** Pia Hauck.

**Project administration:** Pia Hauck.

**Resources:** Pia Hauck.

**Supervision:** Heiko Hecht.

**Validation:** Pia Hauck.

**Visualization:** Pia Hauck.

**Writing – original draft:** Pia Hauck.

**Writing – review & editing:** Pia Hauck, Heiko Hecht.

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
