## [Decision Letter · Decision Letter 0]

16 Aug 2022

PONE-D-22-13721Mood congruence fosters immersion when reading texts while listening to musicPLOS ONE

Dear Dr. Hauck,

Thank you for submitting your manuscript to PLOS ONE. After careful consideration, we feel that it has merit but does not fully meet PLOS ONE’s publication criteria as it currently stands. Therefore, we invite you to submit a revised version of the manuscript that addresses the points raised during the review process.

If you decide to revise the work, please submit a **detailed list of changes for each point** raised when you submit the revised manuscript. Please also highlight **where the text has been changed** in the resubmitted article - this will help to streamline the peer review process and minimise any delays. 

We look forward to receiving your revised manuscript.

Kind regards,

Thiago P. Fernandes, MA, MS, SNeur, PhD

Academic Editor

PLOS ONE

Journal Requirements: 

Reviewers' comments:

Reviewer's Responses to Questions

**Comments to the Author**

1. Is the manuscript technically sound, and do the data support the conclusions?

Reviewer #1: Yes

Reviewer #2: Yes

Reviewer #3: Yes

2. Has the statistical analysis been performed appropriately and rigorously? 

Reviewer #1: Yes

Reviewer #2: Yes

Reviewer #3: No

3. Have the authors made all data underlying the findings in their manuscript fully available?

Reviewer #1: Yes

Reviewer #2: Yes

Reviewer #3: Yes

4. Is the manuscript presented in an intelligible fashion and written in standard English?

Reviewer #1: Yes

Reviewer #2: Yes

Reviewer #3: No

5. Review Comments to the Author

Reviewer #1: Hauck and colleagues in the present study entitled ‘Mood congruence fosters immersion when reading texts while listening to music’ aimed to examine how emotions in texts are related to emotions in the music listened to while reading. For this purpose, two pilot experiments were conducted to assess the conveyed emotional mood of four classical music pieces and that of four narrative text excerpts. In the main experiments, participants had to read the texts while listening to the music and to rate their emotional state in terms of valence and arousal, text and music of the multisensory event in terms of the perceived mood, liking, immersion, and musictext-fit. Results showed a mutual carry-over-effect of happy and sad moods from music to text and vice versa, that was not mediated by the valence, arousal, or dominance experienced by the subject; moreover, they revealed a significant interaction between music mood and text mood.

The main strength of this paper is that it addresses an interesting and timely question, investigating distinct patterns of perceived music-text-fit, demonstrating how the more similar music and text were evaluated in mood, the better the stimuli were regarded to match. In general, I think the idea of this article is really interesting and the authors’ fascinating observations on this timely topic may be of interest to the readers of Plos One. However, some comments, as well as some crucial evidence that should be included to support the authors’ argumentation, need to be addressed to improve the quality of the article, its adequacy, and its readability prior to the publication in the present form. My overall judgment is to publish this article after the authors have carefully considered my suggestions below, in particular reshaping the parts of the Methods and Discussion sections.

Please consider the following comments:

• I suggest changing the title. In my opinion, in the present form it seems to be too wordy and not enough clear and specific.

• Sample: Please provide more information about the tests utilized to assess music expertise and better explain the analysis performed.

• Why did the authors not measure physiological responses of participants? As Rickard (2004) said, physiological parameters can be a truer indicator of emotions rather than methods such as self-report, therefore they could have provided a more reliable measure of music influence on text reading.

• Text stimuli: Please specify if the pilot study was conducted on different participants.

• Also, why the authors did not decide to assess personality traits like anxiety and depression? I believe that having this information would have helped setting a baseline to compare performance between experimental conditions.

• Results: Please specify which post-hoc method was utilized.

• Discussion: In this final section, authors described the results of their study and their argumentation and captured the state of the art well; however, I would have liked to see some views on a way forward. I believe that the authors should make an effort, trying to explain the theoretical implication as well as the translational application of this paper, to adequately convey what they believe is the take-home message of their study. In this regard, I believe that it would be necessary to discuss theoretical and methodological avenues in need of refinement, as well as suggestions of a path forward in understanding inter-relation between emotional appraisals of classical music and narrative text. In this regard, I believe that it could be very interesting to deepen examination of appraisal and how it can influence action tendencies and bahvior: we know that emotion affects behaviors by determining the appropriate action for dealing with or adapting to relevant events both mentally, in the form of states of action readiness or action tendencies, and physically, in the form of physiological responses, and therefore, allowing the monitoring of the potential responses. Recent studies have focused on the power of negative emotions to influence our action control, specifically our inhibitory control (https://doi.org/10.3389/fnbeh.2022.946263;
https://doi.org/10.3389/fpsyg.2018.01334;
https://doi.org/10.3389/neuro.09.013.2008) and heart-related dynamics in human emotional conditioning (learning) in humans (https://doi.org/10.1016/j.tins.2022.04.003;
https://doi.org/10.1037/a0025083;
https://doi.org/10.1111/psyp.14122).

• In my opinion, the ‘Conclusions’ paragraph would benefit from some thoughtful as well as in-depth considerations by the authors, because as it stands, it is very descriptive but not enough theoretical as a discussion should be. Authors should make an effort, trying to explain the theoretical implication as well as the translational application of their research.

• In according to the previous comment, I would ask the authors to better define a ‘Limitations and future directions’ section before the end of the manuscript, in which authors can describe in detail and report all the technical issues that may be brought to the surface.

• Figures: Please, provide higher-quality images because, as it stands, the readers may have difficulty comprehending them. Also, I believe that a figure representing the temporal sequences of experimental phases

Overall, the manuscript contains 3 figures, 6 tables and 96 references. This manuscript might carry important value investigating distinct patterns of perceived music-text-fit, demonstrating how the more similar music and text were evaluated in mood, the better the stimuli were regarded to match.

I hope that, after these careful revisions, the manuscript can meet the Journal’s high standards for publication. I am available for a new round of revision of this article.

I declare no conflict of interest regarding this manuscript.

Best regards,

Reviewer

Reviewer #2: TITLE OF STUDY

Mood congruence fosters immersion when reading texts while listening to music

ABSTRACT OF STUDY

Numerous studies indicate that listening to music and reading are processes that interact in multiple ways. However, these interactions have rarely been explored with regard to the role of emotional mood. In this study, we first conducted two pilot experiments to assess the conveyed emotional mood of four classical music pieces and that of four narrative text excerpts. In the main experiment, participants were asked to read the texts while listening to the music and to rate their emotional state in terms of valence, arousal, and dominance. Subsequently, they rated text and music of the multisensory event in terms of the perceived mood, liking, immersion, and music- text-fit. We found a mutual carry-over-effect of happy and sad moods from music to text and vice versa. Against our expectations, this effect was not mediated by the valence, arousal, or dominance experienced by the subject. Moreover, we revealed a significant interaction between music mood and text mood. Texts were liked better, they were classified as of better quality, and participants felt more immersed in the text if text mood and music mood corresponded. The role of mood congruence when listening to music while reading should not be ignored and deserves further exploration.

GENERAL REMARKS DURING THIS REVIEW:

. This study assessed sensation transfer effects, and multisensory congruency, in the experience of reading text while listening to music. In a combined 2x2 fully crossed within-subjects design, authors observed that both music and reading-text have a mutual influence on the emotional appraisal of the other, suggesting a transfer of emotions. Moreover, both music and text are perceived as more immersing, and are liked better, when music and text mood correspond/fit. A few more interesting and relevant observations are being reported by the authors. Authors also claim that, with this novel study, they are offering a first assessment of cross-modal emotional  mediation involving texts as stimuli.

. I found this study interesting, as well as rather clear to read and understand. However, on the one hand, experimental sample seems rather small for some of the effects being reported (in some cases, n < 10). On the other hand, I am unsure whether this study can be published without ethical approval, since it was conducted in laboratory environment. In fact, PLOSONE clearly specifies that studies involving human subjects should report such an approval. For instance, authors mention in methods sections that "stimuli were considered harmless": So what, and by who? This is usually the role of an ethics committee, since they have the authority and responsibility to make such type of judgements.

SPECIFIC REMARKS DURING THIS REVIEW:

INTRODUCTION:

- PAGE 3, LINE 55: I see the term "sensual" being used in contexts where I would simply use the word "sensory" instead.

- "Our study" subsection, and other parts of manuscript: Why not writing in past, since this study already happened?

METHODS:

- There was just one experimental booth available to everyone?

Reviewer #3: This a very interesting study. The authors are commended for their efforts and well-conducted design and tools. The Title needs some polishing to better describe their findings and variables, avoiding some direct statements. I have some comments that may be helpful.

1 ) Please double check grammar throughout the text

2 ) Double check the references accordingly to the Journal’s standards

3 ) The authors may want to use the STROBE guideline to better detail and refine some sections

4 ) I suggest to avoid the use of vernacular terms not related to the design of the study - it is not experimental, but observational

5 ) My main concern is related to conciseness. Introduction is well written and pretty straightforward, but only to a certain aspect. I highly suggest the authors to scale back and make the intro more concise. Some para.s can be placed as SM, online and, yet, have a refine introduction

6 ) I strongly suggest the use of a detailed section for the eligibility criteria. For example, the participant did not have any disorder (references), neurological condition (references), any other aspect that affects development (references) were infected (covid, for example) etc. Mostly considering some confounding factors can be a hindrance if not controlled

7 ) How to disentangle the impact of confounding factors (e.g. substance use) from the main effect? I mean, can the authors ensure that their outcomes are solely related to the observed variables? What about anxiety, substance use, the use of meds?

8 ) Detail all parameters of your analyses

9 ) Considering this hindrance of the lack of controlling confounding factors, I will make some comments based on what the authors brought in discussion. I think the authors can expand their discussion focusing on important aspects of their entire study and using more than one idea (or reference) per paragraph. It is also important to avoid generalists terms

10 ) I think that the worrying aspect of Discussion is the lack of proper explanation of why did the author interpret the findings. How to explain this relation based on literature? I am not asking the authors to extend or speculate too much, but to give the readers a general impression of the findings and the previous literature

11 ) Also I think the limitations should be placed with recommendations for other authors

12 ) Also I think the strengths need to be better detailed

13 ) A more smoother conclusion is essential

6. PLOS authors have the option to publish the peer review history of their article (what does this mean?). If published, this will include your full peer review and any attached files.

Reviewer #1: No

Reviewer #2: No

Reviewer #3: No

---

## [Author Response · Author response to Decision Letter 0]

31 Oct 2022

R1.1 I suggest changing the title. In my opinion, in the present form it seems to be too wordy and not enough clear and specific.

- Changed it to “Emotionally congruent music and text increase immersion and appraisal“

R1.2 Sample: Please provide more information about the tests utilized to assess music expertise and better explain the analysis performed.

- Added words and sentences to the “Sample” section. The updated paragraph reads as follows: 

“All participants reported being non-synesthetes. Musical expertise was assessed by the Goldsmith Musical Sophistication Index Version 1, a self-report inventory for individual differences in musical sophistication (Gold-MSI; [55]). It measures the ability to engage with music independently of musical style, preferences, or regularity of musical practice. In order to avoid overloading the questionnaire, we picked 18-items assessing the General Factor of Musical Sophistication. In our sample, the mean General Musical Sophistication Score (GMS) was below average compared to the norm (MGMS = 68.97, SDGMS = 23.13; 28th percentile; norm: MGMS = 81.58, SDGMS = 20.62). One participant scored extremely high (GMS = 122); however, her data did not stand out and were thus not dismissed. For analysis, we performed a median split to divide the sample into more (experts) and less (novices) musically experienced participants (medianGMS = 69.25). We identified extreme outlier values in terms of music and text ratings with the help of boxplots (criterion: > 3 * IQR from 1st or 3rd quartile) and replaced these outliers by the group mean. In total, this was the case for five data points representing 0.21 % of all ratings. In addition to the Gold-MSI questionnaire, we had asked for daily music listening and reading routines.” (l. 244 ff.)

R1.3 Why did the authors not measure physiological responses of participants? As Rickard (2004) said, physiological parameters can be a truer indicator of emotions rather than methods such as self-report, therefore they could have provided a more reliable measure of music influence on text reading.

- In our experience, physiological measures did not fare well with regard to valence, and subjective rating have proved more useful for our purposes, which was to assess participants’ subjective appraisal of the presented stimuli. Therefore, we used self-reporting rating scales as independent variables. It would of course be interesting to find out if crossmodal association and transmission can be measured objectively. A follow-up study measuring physiological responses would be an interesting expansion and comparison of subjective with objective measures.

R1.4 Text stimuli: Please specify if the pilot study was conducted on different participants.

- The pilot study was conducted on an independent sample. The corresponding Information was added and the paragraph was changed to:

“In a small pilot experiment with an independent sample (n = 8, 6 female; Mage = 31.38, SDage = 9.47), […]” (l. 295).

R1.5 Also, why the authors did not decide to assess personality traits like anxiety and depression? I believe that having this information would have helped setting a baseline to compare performance between experimental conditions.

- Measuring personality traits like extraversion would indeed have been an interesting addition to the assessed scales, as extraverts have been shown to be less distracted by background music compared to introverts (see l. 701 ff.). However, in this first study to examine this research question, we chose to focus on interpersonal differences in classic variables like age, gender, and expertise due to limited trial length. Additional validated multi-item scales would have overloaded the questionnaire. Also, looking for effects of depression might have required a clinical sample. As our sample consisted of psychology students, we did not expect significant differences in the markedness of depression or anxiety symptoms.

R1.6 Results: Please specify which post-hoc method was utilized.

- We chose to use rmANOVAS as well for our post-hoc analyses, rather than pairwise comparisons. After having identified multivariate effects with the help of rmMANOVAS, we calculated rmANOVAS to check for univariate effects. We were interested in finding out for which within-subject-factor the effect was the strongest. As all within-subjects-factors had only two factor levels, we presumed sphericity of the data and did not apply any correcting factor.

R1.7 Discussion: In this final section, authors described the results of their study and their argumentation and captured the state of the art well; however, I would have liked to see some views on a way forward. I believe that the authors should make an effort, trying to explain the theoretical implication as well as the translational application of this paper, to adequately convey what they believe is the take-home message of their study. In this regard, I believe that it would be necessary to discuss theoretical and methodological avenues in need of refinement, as well as suggestions of a path forward in understanding inter-relation between emotional appraisals of classical music and narrative text. In this regard, I believe that it could be very interesting to deepen examination of appraisal and how it can influence action tendencies and bahvior: we know that emotion affects behaviors by determining the appropriate action for dealing with or adapting to relevant events both mentally, in the form of states of action readiness or action tendencies, and physically, in the form of physiological responses, and therefore, allowing the monitoring of the potential responses. Recent studies have focused on the power of negative emotions to influence our action control, specifically our inhibitory control (https://doi.org/10.3389/fpsyg.2018.01334;
https://doi.org/10.3389/neuro.09.013.2008) and heart-related dynamics in human emotional conditioning (learning) in humans (https://doi.org/10.1016/j.tins.2022.04.003;
https://doi.org/10.1037/a0025083;
https://doi.org/10.1111/psyp.14122).

- This is an interesting point. Our results show that music and text not so much affect the subject’s mood but that the music impinged on the assessment of the text and vice versa. Thus, we are dealing the more or less pure appraisal effects. We could only speculate on how the latter might affect our actions. We have done so by adding the following sentences: 

“Regarding the underlying emotions, we can conclude that music and text did not primarily affect the participants’ mood but rather did the music impinge on the assessment of the text. And vice versa did the text alter the appraisal of the music. Thus, we can assume that we found rather pure appraisal effects here. We may ask how likely such intersensory appraisal changes are to influence our actions. Negative emotions were recently shown to influence our actions, specifically our inhibitory control [64, 65]. Note, however, that the crossmodal appraisal effect in the current study are once removed from the actor’s emotional state. We can thus only speculate if negative emotions identified in a given text or music piece will impact our behavior. Further studies building on this hypothesis would be a promising path forward. “ (l. 586 ff.)

R1.8 In my opinion, the ‘Conclusions’ paragraph would benefit from some thoughtful as well as in-depth considerations by the authors, because as it stands, it is very descriptive but not enough theoretical as a discussion should be. Authors should make an effort, trying to explain the theoretical implication as well as the translational application of their research.

- We see your point but are reluctant to go beyond our findings in the conclusion. We have done so in the discussion. In our view, the Conclusions section is to summarize the most important aspects in a digestible way, as to provide the most relevant findings and applications of the research. In-Depth considerations and theoretical explanations are given in the previous sub-sections of the Discussion. However, we added one sentence about emotion mediation to the Conclusion section as follows:

“These results were, however, not mediated by the valence, arousal, or dominance experienced by the participant, which makes this research stand out from the majority of crossmodal studies.” (l. 753 ff.)

R1.9 In according to the previous comment, I would ask the authors to better define a ‘Limitations and future directions’ section before the end of the manuscript, in which authors can describe in detail and report all the technical issues that may be brought to the surface.

- We added the following paragraph to the ‘Limitations and future directions’ section to address more technical issues:

“On a more technical level, authors interested in conducting studies including musical and reading stimuli, should take into account that different reading speed of participants leads to differing combinations of text passages with musical phrases. In our study, we addressed this issue by cutting and looping relatively uniform music excerpts excluding more important changes in loudness, tempo, instrumentation, or mode. A different approach could be to control the reading speed by presenting single sentences on a screen, or using monotonous sound bites rather than music to reduce variance in the auditory domain. However, this was not suitable for our study as we wanted to create a reading and listening situation that is as realistic as possible in a laboratory setting. Further, the question of the auditory delivery is crucial to any crossmodal study design including auditory stimuli. We chose over-ear headphones to approximate everyday reading-while-listening situations and to prevent distraction through external noise. Nevertheless, the influence of music transmitted through loudspeakers and thereby influencing the overall room ambiance would be interesting to examine.” (l. 711 -724)

R1.10 Figures: Please, provide higher-quality images because, as it stands, the readers may have difficulty comprehending them. Also, I believe that a figure representing the temporal sequences of experimental phases

- Figures are now provided in higher quality, being checked on the PLOS one portal PACE.

- A figure showing the sequence of experimental blocks was added as Figure 2.

Reviewer #2:

R2.1 I found this study interesting, as well as rather clear to read and understand. However, on the one hand, experimental sample seems rather small for some of the effects being reported (in some cases, n < 10). 

- In line with many other studies in the field, in our view, the sample size of n = 40 is well suited for experimental study designs examining crossmodal correspondences. We do agree that some ex-post sub-groups of the 40 can only be looked at exploratively. For the tentative effects reported for subgroups with n < 10, we now explicitly mention this caveat: 

“As the group of those being used to simultaneous reading while and listening to music was very small (n = 7), we could only exploratively examine if habituation might be an influencing factor for the crosstalk between music and text.” (l. 267 ff.)

R2.2 On the other hand, I am unsure whether this study can be published without ethical approval, since it was conducted in laboratory environment. In fact, PLOSONE clearly specifies that studies involving human subjects should report such an approval. For instance, authors mention in methods sections that "stimuli were considered harmless": So what, and by who? This is usually the role of an ethics committee, since they have the authority and responsibility to make such type of judgements.

- We have operated under a blanket approval of our local ethics committee. Thus, everything should be with PlosOne guidelines. We have made this explicit now, the section was changed to: 

“The experiment was conducted in accordance with the Declaration of Helsinki. Informed consent was obtained before the experimental session, subjects were told that they could abort the experiment at any time without consequences, and they were debriefed about the purposes of the study immediately after the session. The study was approved by a local Departmental Ethics committee under a blanket approval.” (l. 269 ff.)

SPECIFIC REMARKS DURING THIS REVIEW:

INTRODUCTION:

R2.3 PAGE 3, LINE 55: I see the term "sensual" being used in contexts where I would simply use the word "sensory" instead.

- Thanks for catching this we have changed accordingly (now l. 54)

R2.4 "Our study" subsection, and other parts of manuscript: Why not writing in past, since this study already happened?

- We have changed to past tense where appropriate (l. 218, 222, 224)

METHODS:

R2.5 There was just one experimental booth available to everyone?

- Yes, there was one booth, but participants were tested separately. The only other person being present throughout the experimental sessions was the investigator. 

Reviewer #3: This a very interesting study. The authors are commended for their efforts and well-conducted design and tools. 

The Title needs some polishing to better describe their findings and variables, avoiding some direct statements. 

- Please consider our alternative suggestion: “Emotionally congruent music and text increase immersion and appraisal”

I have some comments that may be helpful.

R3.1 Please double check grammar throughout the text

- We checked the grammar and hope to have caught the rough spots.

R3.2 Double check the references accordingly to the Journal’s standards

- We checked the references and made changes where necessary.

R3.3 The authors may want to use the STROBE guideline to better detail and refine some sections

- We added missing methods detail (see answers to Reviewer 1)

R3.4 I suggest to avoid the use of vernacular terms not related to the design of the study - it is not experimental, but observational

- We hope to have clarified in the revision where we share observations and where we report experimental results and conclusions. If not, please let us know the objectionable terms.

R3.5 My main concern is related to conciseness. Introduction is well written and pretty straightforward, but only to a certain aspect. I highly suggest the authors to scale back and make the intro more concise. Some para.s can be placed as SM, online and, yet, have a refine introduction

- We agree that our introduction reviews the field thoroughly. We also believe it is a matter of taste how much one likes to offer here. The reader may skip parts if familiar with the topic, for those unfamiliar, it might be useful. So, we would like to keep the introduction complete.

R3.6 I strongly suggest the use of a detailed section for the eligibility criteria. For example, the participant did not have any disorder (references), neurological condition (references), any other aspect that affects development (references) were infected (covid, for example) etc. Mostly considering some confounding factors can be a hindrance if not controlled

- The study was conducted not within a clinical setting, but within a university setting with students as participants. We did require that they were of normal, subjectively healthy eyesight and hearing, but did not explicitly ask for specific disorders, neurological loss or other compounding health conditions. We agree that this should be considered for future designs. To be frank about this, we have added this to the text as follows:

“Potentially confounding factors such as psychological and neurological disorders and current infections were not assessed, which might be recommended for future samples.” (l. 692)

R3.7 How to disentangle the impact of confounding factors (e.g. substance use) from the main effect? I mean, can the authors ensure that their outcomes are solely related to the observed variables? What about anxiety, substance use, the use of meds?

- Note that we had a fully crossed within-subjects design. Any clinical issue – if present - would be be present equally in all conditions. We cannot rule out that say a cannabis using subject was differently affected by the music than were others, but this would have been so in all experimental conditions. Regarding anxiety and other individual differences, they would of course be interesting to explore in follow-up studies.

R3.8 Detail all parameters of your analyses

- All parameters are either displayed in the results section or in the tables placed in the appendix.

R3.9 Considering this hindrance of the lack of controlling confounding factors, I will make some comments based on what the authors brought in discussion. I think the authors can expand their discussion focusing on important aspects of their entire study and using more than one idea (or reference) per paragraph. It is also important to avoid generalists terms

R3.10 I think that the worrying aspect of Discussion is the lack of proper explanation of why did the author interpret the findings. How to explain this relation based on literature? I am not asking the authors to extend or speculate too much, but to give the readers a general impression of the findings and the previous literature

- Answer to R3.9 and R3.10: We have now emphasized that emotional appraisal rather than changes of emotional state are at the center of our findings. Each of the paragraphs in the discussion section contains at least one possible explanation for our findings and references to literature. We do not want to engage in too much speculation and thus focused on the most promising ideas.

R3.11 Also I think the limitations should be placed with recommendations for other authors

- Thank you for this suggestion. We have extended the ‘Limitations and future directions’ section: 

“On a more technical level, authors interested in conducting studies including musical and reading stimuli, should take into account that different reading speed of participants leads to differing combinations of text passages with musical phrases. In our study, we addressed this issue by cutting and looping relatively uniform music excerpts excluding more important changes in loudness, tempo, instrumentation, or mode. A different approach could be to control the reading speed by presenting single sentences on a screen or using monotonous sound bites rather than music to reduce variance in the auditory domain. However, this was not suitable for our study as we wanted to create a reading and listening situation that is as realistic as possible in a laboratory setting. Further, the question of the auditory delivery is crucial to any crossmodal study design including auditory stimuli. We chose over-ear headphones to approximate everyday reading-while-listening situations and to prevent distraction through external noise. Nevertheless, the influence of music transmitted through loudspeakers and thereby influencing the overall room ambiance would be interesting to examine.” (l. 711 -724)

R3.12 Also I think the strengths need to be better detailed

- We have added the following statement:

“Despite of the described limitations, the present study is of great value and an important addition to the field. Its strengths include its novelty in terms of stimulus combinations, its applicability, and its timely relevance. The design is well-balanced between practical relevance and control of conditions and the results can be regarded as a solid base for further investigations.” (l. 725)

R3.13 A more smoother conclusion is essential

- We have reworked the conclusion.

---

## [Decision Letter · Decision Letter 1]

6 Nov 2022

PONE-D-22-13721R1Emotionally congruent music and text increase immersion and appraisalPLOS ONE

Dear Dr. Hauck,

Thank you for submitting your very good work!You will notice that reviewers’ comments are good and you satisfied most of their concerns.Based on my own reading, I ask you to give more details on eligibility, stats and a smooth conclusion with further directions for researchers and readers. Also, maybe the authors want to make some parts available on OSF or so, then others will easily replicate your study.It’s basically a few and quick amendments to be made, but nothing worrying. I will be happy to receive your edits and proceed with your study.From now, I am anticipating my sincere “good luck” with this very good one and relevant work.Hope the authors can consider PLOS again.

We look forward to receiving your revised manuscript.

Kind regards,

Thiago P. Fernandes, PhD

Academic Editor

PLOS ONE

Journal Requirements:

Additional Editor Comments:

Please read my comments

Reviewers' comments:

Reviewer's Responses to Questions

**Comments to the Author**

1. If the authors have adequately addressed your comments raised in a previous round of review and you feel that this manuscript is now acceptable for publication, you may indicate that here to bypass the “Comments to the Author” section, enter your conflict of interest statement in the “Confidential to Editor” section, and submit your "Accept" recommendation.

Reviewer #1: All comments have been addressed

Reviewer #3: (No Response)

2. Is the manuscript technically sound, and do the data support the conclusions?

Reviewer #1: Yes

Reviewer #3: Yes

3. Has the statistical analysis been performed appropriately and rigorously? 

Reviewer #1: Yes

Reviewer #3: Yes

4. Have the authors made all data underlying the findings in their manuscript fully available?

Reviewer #1: Yes

Reviewer #3: Yes

5. Is the manuscript presented in an intelligible fashion and written in standard English?

Reviewer #1: Yes

Reviewer #3: Yes

6. Review Comments to the Author

Reviewer #1: Manuscript ID: PONE-D-22-13721_R1 2nd Round

The authors did an excellent job clarifying all the questions I have raised in my previous round of review. Currently, this paper entitled ‘Emotionally congruent music and text increase immersion and appraisal’, is a well-written, timely piece of research that examined how emotions in texts are related to emotions in the music listened to while reading.

Overall, this is a timely and needed work. It is well researched and nicely written, and describes in detail distinct patterns of perceived music-text-fit, demonstrating how the more similar music and text were evaluated in mood, the better the stimuli were regarded to match.

I believe that this paper does not need a further revision, therefore the manuscript meets the Journal’s high standards for publication.

I am always available for other reviews of such interesting and important articles.

Thank You for your work.

Reviewer #3: Thank you for your very diligent and careful work. I only have a few suggestions, but nothing time consuming:

> Regarding the Methods, more details are need. For example, the authors need to explain the timeline, screening, confounding variables, the Cronbach values for any scale, all details of the analyses. More specifically, details on eligibility is needed: were the patients free of cognitive disorders? (10.3389/fpsyt.2013.00182), had normal visual acuity (10.1016/j.jpsychires.2022.03.014 and 10.1016/j.clinph.2005.06.013), had no previous contact with substances (10.3389/fpsyg.2018.00288), if they had previous contact with any other perceptual task before or were naive to them;

> Use more illustrative graphs to demonstrate your findings;

> Use effect sizes and CIs for the analysis;

> Use the corrected p-value;

> Boxplots with individuals values are more representative of the data and the sample (10.1038/nmeth.2813)

7. PLOS authors have the option to publish the peer review history of their article (what does this mean?). If published, this will include your full peer review and any attached files.

Reviewer #1: No

Reviewer #3: No

---

## [Author Response · Author response to Decision Letter 1]

18 Dec 2022

PONE-D-22-13721R1 – Response to reviews

E.1 eligibility, stats and a smooth conclusion with further directions for researchers and readers. 

- We have worked on the conclusion and put an emphasis on future directions. 

E.2 Also, maybe the authors want to make some parts available on OSF or so, 

- We uploaded the lists of stimuli, the questionnaires, and all data here: DOI 10.17605/OSF.IO/SF47R. 

- The specific texts and music files used as stimuli could not be uploaded due to copyright regulations.

E.3 - We’ve done it accordingly, see Cover Letter

E.4 Please review your reference list to ensure that it is complete and correct. If you have cited papers that have been retracted, please include the rationale for doing so in the manuscript text, or remove these references and replace them with relevant current references. Any changes to the reference list should be mentioned in the rebuttal letter that accompanies your revised manuscript. If you need to cite a retracted article, indicate the article’s retracted status in the References list and also include a citation and full reference for the retraction notice.

- We have reviewed the reference list again, but we did not find any papers that have been retracted. Is there a particular paper you have in mind? Maybe the issue is with the dates on e-publication ahead of the print issue? We have updated to print issue citation where available. Further, of the references in ‘submitted’ status had been published in the meantime. We also made some changes to journal titles, now using short titles.

- We also corrected the numbering of tables.

R3.1 Regarding the Methods, more details are need. For example, the authors need to explain the timeline, screening, confounding variables, the Cronbach values for any scale, all details of the analyses.

- We added one sentence about the timeframe in which data were gathered. (l. 374)

- All details of the analyses can be found in the Supplementary Material.

R3.2 More specifically, details on eligibility is needed: were the patients free of cognitive disorders? (10.3389/fpsyt.2013.00182), 

- We cannot be 100 % certain. We added a sentence about the non-performance of cognitive tests (l. 248-250)

R3.3 had normal visual acuity (10.1016/j.jpsychires.2022.03.014 and 10.1016/j.clinph.2005.06.013), 

- We relied on self-reports as to visual and hearing acuity. We added the word “self-reported” to the respective sentence (see l. 244)

R3.4 had no previous contact with substances (10.3389/fpsyg.2018.00288), 

- We added a sentence about the non-testing of substance use. (l. 248-250) In our understanding, it is not highly relevant for this kind of experiment on multisensory perception, as it does not include any clinical trials.

R3.5 if they had previous contact with any other perceptual task before or were naive to them;

- We added two sentences about this, see l. 240-243

R3.6 > Use more illustrative graphs to demonstrate your findings;

- After having thoroughly evaluated different kinds of graphics to further illustrate our findings, we concluded that the existing three graphs in conjunction with tables are the most comprehensible and simple way to show our findings. Nothing would be gained but some information lost if we were to replace the tables by graphics. 

R3.7 > Use effect sizes and CIs for the analysis; Use the corrected p-value;

- Effect sizes (η²) are included in the results tables (see Tables 1, 2, S4 – S8). p-values as results of post-hoc analyses were LSD-corrected (we added this information to the text in l. 395). Other corrections were not indicated.

R3.8 > Boxplots with individuals values are more representative of the data and the sample (10.1038/nmeth.2813)

- Yes, agreed. As this would bloat the manuscript, we added boxplots to the supplementary materials (see S1 Figure) and refer to them at the beginning of the results section (see l. 386-378).

---

## [Editor Report · Decision Letter 2]

21 Dec 2022

Emotionally congruent music and text increase immersion and appraisal

PONE-D-22-13721R2

Dear Dr. Hauck,

Thank you for your politeness and thoughtful edits.

After re-reading, I am convinced that all concerns were addressed.

We’re pleased to inform you that your manuscript has been judged scientifically suitable for publication and will be formally accepted for publication once it meets all outstanding technical requirements.

Kind regards,

Thiago P. Fernandes, PhD

Academic Editor

PLOS ONE

Additional Editor Comments (optional):

Thank you for your valuable submission.

---

## [Editor Report · Acceptance letter]

4 Jan 2023

PONE-D-22-13721R2 

Emotionally congruent music and text increase immersion and appraisal 

Dear Dr. Hauck:

I'm pleased to inform you that your manuscript has been deemed suitable for publication in PLOS ONE. Congratulations! Your manuscript is now with our production department. 

Kind regards, 

on behalf of

Dr. Thiago P. Fernandes 

Academic Editor

PLOS ONE